# Clinical Evaluation of a Line-Probe Assay for Tuberculosis Detection and Drug-Resistance Prediction in Namibia

G. Günther,[a,b,c] E. Saathoff,[d,e] A. Rachow,[d,e] H. Ekandjo,[b] A. Diergaardt,[b] N. Marais,[b] C. Lange,[f,g,h,i] E. Nepolo[b]

[a]Department of Pulmonology and Allergology, Inselspital, Bern University Hospital, University of Bern, Bern, Switzerland

[b]Department of Human, Biological and Translational Medical Sciences, School of Medicine, University of Namibia, Windhoek, Namibia

[c]Department of Medicine, Katutura State Hospital, Windhoek, Namibia

[d]Division of Infectious Diseases and Tropical Medicine, University Hospital, LMU Munich, Munich, Germany

[e]German Centre for Infection Research (DZIF), Partner Site Munich, Germany

[f]Research Center Borstel, Leibniz Lung Center, Borstel, Germany

[g]Tuberculosis Unit, German Center for Infection Research (DZIF), Borstel, Germany

[h]Respiratory Medicine & International Health, University Lübeck, Lübeck, Germany

[i]Baylor College of Medicine and Texas Children's Hospital, Global TB Program, Houston, Texas, USA

C. Lange and E. Nepolo contributed equally to this article.

**ABSTRACT** Treatment of tuberculosis requires rapid information about *Mycobacterium tuberculosis* (*Mtb*) drug susceptibility to ensure effective therapy and optimal outcomes. At the tuberculosis referral hospital in Windhoek, Namibia, a country of high tuberculosis incidence, we evaluated the diagnostic accuracy of a line-probe-assay (LPA), GenID, for the molecular diagnosis of *Mtb* infection and drug resistance in patients with suspected tuberculosis (cohort 1) and confirmed rifampin (RIF)-resistant tuberculosis (cohort 2). GenID test results were compared to Xpert MTB/RIF and/or *Mtb* culture and antimicrobial suceptibilty testing. GenID LPA was applied to 79 and 55 samples from patients in cohort 1 and cohort 2, respectively. The overall sensitivity of GenID LPA for the detection of *Mtb* DNA in sputum from patients with detectable and undetectable acid-fast bacilli by sputum smear microscopy was 93.3% (56/60; 95% confidence interval = 83.8–98.2) and 22.7% (5/22; 7.8–45.4). The sensitivity/specificity for the detection of drug resistance was 84.2% (32/38; 68.7–94.0)/100% (19/19; 82.4–100.0) for RIF, 89.7% (26/29; 72.6–97.8)/91.7% (22/24; 73.0–99.0) for isoniazid, and 85.7% (6/7; 42.1–99.6)/94.7% (18/19; 74.0–99.9) for fluoroquinolones; 23.6% of tests for second-line injectable resistance were invalid despite repeat testing. The diagnosis of tuberculosis by detection of *Mtb* DNA in sputum by GenID LPA depends strongly on the detection of acid-fast bacilli in sputum specimen. Prediction of drug resistance by GenID did not reach the World Health Organization (WHO) target product profile.

**IMPORTANCE** *Mycobacterium tuberculosis* (*Mtb*) drug-resistance detection is crucial for successful control of tuberculosis. Line-probe assays (LPA) are frequently used to detect resistance to rifampin, isoniazid, fluoroquinolones (FQs), and second-line injectables (SLIs). GenID RIF/isoniazid (INH), FQ, and SLI LPA have not been widely tested and used so far. This study tested the diagnostic performance of the GenID LPA in a high-incidence TB/HIV, real-world setting in Namibia. The LPA demonstrates only an acceptable diagnostic performance for *Mtb* and drug-resistance detection. The diagnostic sensitivity and specificity fall short of the WHO suggested target product profiles for LPA.

**KEYWORDS** Xpert MTB/RIF, mycobacterium tuberculosis detection, second-line resistance, mycobacterium tuberculosis, diagnostics, multi-drug resistant

Address correspondence to G. Günther, gunar.guenther@insel.ch.

The authors declare no conflict of interest.

Tuberculosis (TB) and drug-resistant TB (DR-TB) remain major global health challenges (1). Easy to use and accurate diagnostics for TB and drug resistance are essential to reduce the global burden of TB. Nucleic acid amplification tests (NAATs) like Xpert MTB/RIF and Xpert MTB/RIF Ultra allow highly sensitive and specific detection of *Mycobacterium tuberculosis* (*Mtb*) in sputum specimen and are the first-line test for TB diagnosis in Namibia since 2017. Sputum-based line-probe-assays (LPA) complement rapid molecular TB drug-resistance testing, mostly by detection of resistance-associated mutations in *Mtb* against rifampin (RIF), isoniazid (INH), fluoroquinolones (FQ), second-line injectable (SLI) drugs (i.e., amikacin [AM], capreomycin [CM], streptomycin [STR], and kanamycin [KM]), and ethambutol (EMB). Commercially available LPAs, which are recommended by the World Health Organization (WHO) are as follows: INNO-LiPA Rif.TB assay (Innogenetics, Ghent, Belgium), Genotype MTBDR*plus* assay and MTBDR*sl* assay (Hain Lifescience, Nehren, Germany), and Nipro NTM+MDRTB assay (Nipro, Tokyo, Japan). Currently, WHO recommends the use of LPAs in patients with detectable acid-fast bacilli (AFB) and from culture-positive specimens as initial test for detection of resistance to RIF and INH instead of culture-based phenotypic antimicrobial susceptibility testing (AST). In addition, LPAs are recommended in patients with multidrug-resistant (MDR)/RIF-resistant (RR) TB as initial test to detect resistance to FQ and SLI (2).

Based on NAAT and *in situ* hybridization technology, LPAs provide valuable information about DR-TB in about 36–48 h after specimen collection (3). LPA diagnostic performance for *Mtb* detection is mostly compared with *Mtb* culture (4). AST performance is compared increasingly to molecular (i.e., next-generation genome sequencing) and culture-based methods as reference standards (5).

GenID LPA (AID Autoimmun Diagnostika GmbH, Strassberg, Germany) is a commercially available LPA, which is based on the technology of NAAT and *in situ* hybridization. The LPA can be used for the detection of *Mtb* complex and *Mtb* drug-resistance mutations against RIF, INH, EMB, FQ, and SLI.

The GenID RIF/INH module identifies INH resistance if there are mutations at the −16, −15, and −8 *inhA* or the S315T KatG amino acid locus and RIF resistance, if there are mutations in the codons 516, 526, and 531 of the *rpoB* gene. The GenID SLI strip documents resistance to STR, AM, KM, and CM. Resistance to STR is defined by mutations in position 43 and 88 of RpsL gene and at position 513, 514, 515, and 517 of the *rrs* gene. Mutations at positions rrs 1401 and 1402 are associated with resistance to AM/KM and mutations at position rrs 1484 with AM/KM/CM resistance (6). The GenID FQ detects mutations at positions 90, 91, and 94 of gyrA, while EMB resistance is assumed if mutations in positions 306 and 918 of embB gene occur.

So far, GenID LPA has only been evaluated in high-technology laboratories in Europe, mostly based on culture isolates, but not in a limited resource setting with high-TB incidence and a high proportion of TB/HIV coinfection (7, 8).

Namibia has a high burden of TB with an incidence rate of 460 (328–614) per 100,000 inhabitants in 2020. The estimated number of annual new cases is 12,000 (8,300–16,000), about 30% coinfected with HIV (1, 9, 10). MDR/RR-TB is present in 4.5% of new cases and 7.9% of previously treated cases (11). Xpert MTB/RIF (Cepheid, Sunnyvale, CA, USA) was introduced in 2017 (just before the start of this study) as a first-line TB diagnostic test in addition to sputum smear microscopy. TB diagnosis is guided by a national algorithm (12), where culture is routinely only performed if RR-TB is detected by Xpert MTB/RIF and in extrapulmonary and pediatric specimens.

We evaluate the diagnostic performance of GenID LPA in a prospective cohort of patients with a suspected diagnosis of TB and a second prospective cohort of patients with RR-TB, at a TB referral center in Windhoek, Namibia.

## RESULTS

**Description of cohorts.** Seventy-nine of 87 (90.8%) patients with a presumptive diagnosis of TB (cohort 1) and 55/67 (83.3%) patients with a confirmed diagnosis of RR-

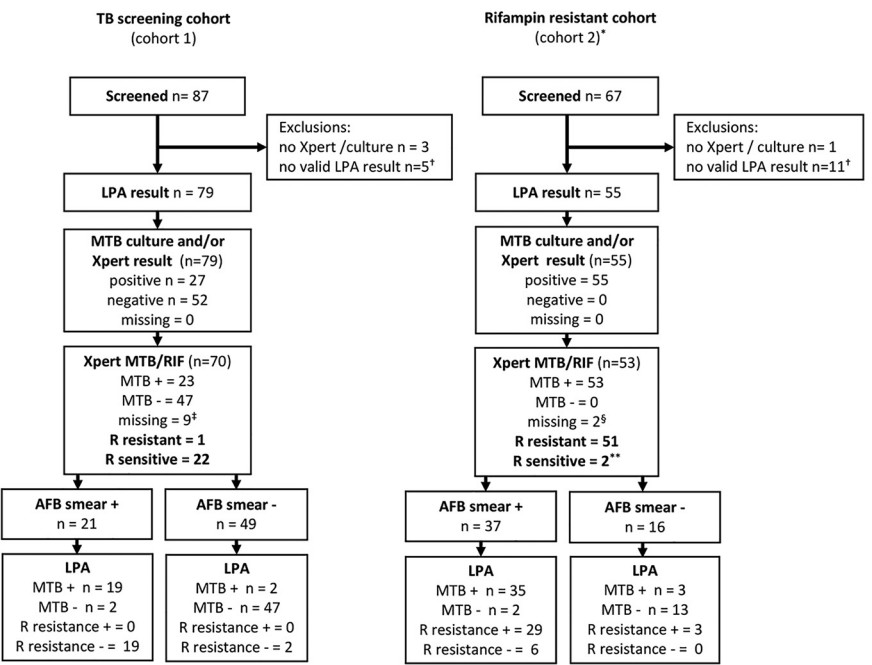

**FIG 1** Flow chart of samples included in tuberculosis (TB) screening cohort 1 and rifampin-resistant cohort 2 and line-probe-assay (LPA) testing results, stratified by sputum smear microscopy. MTB, *Mycobacterium tuberculosis*; RIF, rifampin; AFB, acid-fast bacilli; *, all participants were included in the rifampin- resistant treatment cohort, based on samples with Xpert MTB/RIF or DST and started rifampin- resistant TB treatment and samples were referred for LPA; †, no valid result for MTB detection with GenID LPA RIF/INH strip; ‡, samples from 70 participants in cohort 1 did receive a Xpert MTB/RIF test, 23 were Xpert MTB/RIF positive, 4 *Mtb* culture positive. Missing Xpert MTB/RIF results are related to laboratory stock out. No routine *Mtb* cultures are done for samples from Xpert MTB/RIF negative patients under programmatic conditions in Namibia, 4 cases were detected by positive *Mtb* culture, which was done for study purposes; §, 2 samples from participants in cohort 2 had no Xpert MTB/RIF result, but a positive *Mtb* culture; **, 2 samples from participants in cohort 2 were on Xpert MTB/RIF rifampin sensitive, but on phenotypic AST samples resistant to rifampin MTB, LPA, and AFB.

TB (cohort 2) had a valid result for *Mtb* DNA detection in sputum specimen using GenID RIF/INH strip and were included in the final analysis (Fig. 1). Eight patients in cohort 1 (no Xpert/culture, *n* = 3; no valid LPA, *n* = 5) and 12 in cohort 2 (no Xpert/culture, *n* = 1; no valid LPA, *n* = 11) were excluded. In cohort 1, 50/79 (73.5%) of the patients with an available HIV test result were HIV positive. The prevalence of HIV infection in cohort 2 was 26/53 (49.1%) (Table 1).

**TABLE 1** Cohort characteristics[a]

| Characteristic | Cohort 1 (*n* = 79)[b] | Cohort 2 (*n* = 55)[b] |
|---|---|---|
| Male, *n* (%) | 45 (57.0) | 36 (65.5) |
| Age, median (IQR) | 41.3 (31.8–48.2) | 34.5 (27.7–42.8) |
| Weight, median (IQR) (kg) | 52.0 (45.5–58.0) | 48.3 (41.8–56.7) |
| Previous TB treatment, *n* (%) | 27 (34.6) | 28 (50.9) |
| HIV positive, *n* (%) | 50 (73.5) | 26 (49.1) |
| HIV results missing, *n* (%) | 11 | 2 |
| If HIV–on ART, *n* (%) | 42 (84.0) | 20 (76.9) |
| Haemoglobin at diagnosis, median (IQR) (g/dL) | 10.9 (8.3–12.3) | 11.6 (9.7–13.2) |
| No. of persons in household, median (IQR) | 5.0 (3.0–7.0) | 5.0 (3.0–7.0) |
| Alcohol intake, *n* (%) | | |
| None | 54 (69.2) | 32 (58.2) |
| Moderate | 24 (30.8) | 23 (41.8) |
| Excessive | 0 | 0 |
| No information | 1 | 0 |
| Active cigarette smoking yes, *n* (%) | 5 (6.3) | 6 (10.9) |

[a]TB, tuberculosis; IQR, interquartile range.
[b]All patients with line-probe-assay result.

In cohort 1, 27/79 (34.2%) patients with symptoms of TB were tested positive for *Mtb* by Xpert MTB/RIF or culture. In cohort 2, RIF resistance was confirmed by Xpert MTB/RIF or AST for all 55 (100.0%) patients with a valid result for *Mtb* DNA detection on LPA.

**M. tuberculosis detection.** When combining both cohorts (Table 2), the overall sensitivity and specificity of the LPA RIF/INH test for *Mtb* detection was 74.4% (61/82; 95% confidence interval = 63.6–83.4) and 98.1% (51/52; 89.7–100.0), respectively. Table S1 in the supplemental material documents a LPA sensitivity for cohort 1 of 81.5% (22/27; 61.9–93.7) and for cohort 2 of 70.9% (39/55; 57.1–82.4) and a specificity of 98.1% (51/52; 89.7–100.0) in cohort 1. Cohort 2 had no patients without TB diagnosis, so specificity could not be assessed.

LPA sensitivity was higher in patients with detectable AFB (93.3%; 56/60; 83.8–98.2) than in patients with undetectable AFB on sputum smear microscopy (22.7%; 5/22; 7.8–45.4; $P < 0.0001$). Similar results were observed in a separate analysis of cohort 1 and 2 (Table S1).

Detection of *Mtb* DNA was reduced in HIV-positive (66.7%, 51.0-80.0) compared to HIV-negative patients (87.9%; 71.8–9 6.6; $P = 0.0311$).

**Detection of mutations predicting M. tuberculosis drug resistance.** In cohort 2, 29/37 (78.4%) patients with detectable AFB on sputum smear microscopy also had RIF resistance detected by LPA, while only 3/16 (19%) of patients with undetectable AFB on smear microscopy had RIF resistance detected by LPA (Fig. 1).

The overall diagnostic sensitivity of the LPA for RIF and INH-resistance detection in both cohorts was 84.2% (32/38; 68.7–94.0) and 89.7% (26/29; 72.6–97.8), respectively, while specificity was 100% (19/19; 82.4–100) and 91.7% (22/24; 73.0–99.0), respectively. In cohort 1 with presumed TB, one single patient had RIF resistance in Xpert MTB/RIF, but resistance was not confirmed by LPA and AST. Table 2 also shows the diagnostic sensitivities in HIV-positive versus HIV-negative patients, while Table S1 shows diagnostics sensitivities and specificities for RIF and INH testing separate for cohort 1 and 2.

Second-line drug resistance was tested by LPA for all samples from cohort 2 (Table 3). FQ resistance was detected with 85.7% (6/7; 42.1–99.6) sensitivity and 94.7% (18/19; 74.0–99.9) specificity and SLI resistance with 75.0% (3/4; 19.4–99.4) sensitivity and 87.5% (7/8; 47.3–99.7) specificity.

**Repeated LPA testing.** Forty-two out of one-hundred thirty-four (31.8%) patient samples in cohort 1 and cohort 2, evaluated with the LPA RIF/INH test for *Mtb* detection, required repeated testing in order obtain a valid test result. Twenty-nine out of fifty-four (53.7%) and 19/42 (45.2%) of valid LPA tests for FQ and SLI drug resistance were obtained by repeat testing. Despite repeated testing, no interpretable results were obtained in 1/55 (1.8%) and in 13/55 (23.6%) LPA tests for FQ and SLI drug resistance, respectively (Table 4).

**Frequency of mutations.** Table S2 shows the distribution of drug-resistance-associated mutations identified by LPA.

## DISCUSSION

We evaluated the diagnostic performance of GenID, a LPA for the rapid detection of TB and *Mtb* drug resistance in two cohorts of patients in Namibia, a country with a high burden of TB and HIV. We found that detection of *Mtb* DNA with GenID LPA as a surrogate marker for active TB is dependent on a sufficient amount of bacillary DNA in the sputum sample. While the test sensitivity was 93% in patients with detectable AFB on sputum smear microscopy, test sensitivity was only 23% in patients without AFB in sputum smear microscopy. In addition, the test performance was reduced by 6% of samples in cohort 1 and 16% in cohort 2, which had to be excluded from the analysis mostly as no valid LPA result for the GenID RIF/INH strip could be produced. The sensitivity to detect RIF, INH, FQ, and SLI resistance was for all tested drugs less than 90%, while WHO target product profiles was set at optimal above 95% for RIF, FQ, PZA, and INH, if culture-based AST is the reference standard (13). Almost one-quarter of assays results for SLI drug resistance remained inconclusive despite repeated testing.

Two previous studies evaluating GenID LPA analyzed almost exclusively samples from patients with detectable AFB on sputum smear microscopy (7, 8). We found insufficient

**TABLE 2** Sensitivity and specificity of GenID LPA RIF/INH for detection of TB, compared with MGIT culture and/or Xpert MTB/RIF; for detection of RIF resistance, compared to Xpert MTB/RIF; and for INH resistance, compared to INH culture/AST[a]

| GenID RIF/INH module | All (n = 134) | | HIV+ (n = 76)[b] | | HIV− (n = 45)[b] | |
|---|---|---|---|---|---|---|
| | Sensitivity | Specificity | Sensitivity | Specificity | Sensitivity | Specificity |
| Mtb result all patients[c] | 61/82 = 74.4% (63.6–83.4) | 51/52 = 98.1% (89.7–100.0) | 30/45 = 66.7% (51.0–80.0) | 30/31 = 96.8% (83.3–99.9) | 29/33 = 87.9% (71.8–96.6) | 12/12 = 100% (73.5–100) |
| Mtb result among smear positive[c] | 56/60 = 93.3% (83.8–98.2) | 0/1 = 0.0% (0.0–97.5) | 27/30 = 90.0% (73.5–97.5) | 0/1 = 0.0% (0.0–97.5) | 27/27 = 100% (87.2–100) | n/a |
| Mtb result among smear negative[c] | 5/22 = 22.7% (7.8–45.4) | 51/51 = 100% (93.0–100) | 3/15 = 20.0% (4.3–48.1) | 30/30 = 100% (88.4–100) | 2/6 = 33.3% (4.3–77.7) | 12/12 = 100% (73.5–100) |
| RIF resistance among Mtb+[d,e] | 32/38 = 84.2% (68.7–94.0) | 19/19 = 100% (82.4–100) | 11/14 = 78.6% (49.2–95.3) | 14/14 = 100% (76.8–100) | 21/23 = 91.3% (72.0–98.9) | 5/5 = 100% (47.8–100) |
| INH resistance among Mtb+[d,e] | 26/29 = 89.7% (72.6–97.8) | 22/24 = 91.7% (73.0–99.0) | 8/9 = 88.9% (51.8–99.7) | 17/19 = 89.5% (66.9–98.7) | 18/20 = 90.0% (68.3–98.8) | 4/4 = 100% (39.8–100) |

[a]Analysis was performed for all patients from both cohorts. See Table S1 for the separate diagnostic performance for cohort 1 and 2. AST, antimicrobial susceptibility testing by culture; Mtb, Mycobacterium tuberculosis; INH, isoniazid; RIF, rifampin.
[b]11 patients in cohort 1 and 2 patients in cohort 2 did not receive an HIV test.
[c]Mtb detection is based on Xpert MTB/RIF and/or culture (Bactec MGIT 960) as reference standard, and Mtb detection of LPA was assessed in the GenID RIF/INH module.
[d]RIF resistance was declared when rpoB mutation present or rpoB wild type was missing, and INH resistance was declared when inhA and/or katG mutation was present or inhA and/or katG wild type was missing.
[e]RIF resistance is based on Xpert MTB/RIF as reference standard, and INH resistance is based on culture/AST as reference standard.

**TABLE 3** Sensitivity and specificity of GenID LPA in cohort 2 for FQ, SLI, EMB, and STR resistance detection, compared with MGIT culture/AST[a]

| GenID FQ and SLI module | Sensitivity | Specificity |
|---|---|---|
| FQ resistance[b] | 6/7 = 85.7% (42.1–99.6) | 18/19 = 94.7% (74.0–99.9) |
| SLI resistance[c] | 3/4 = 75.0% (19.4–99.4) | 7/8 = 87.5% (47.3–99.7) |
| EMB[d] resistance | 11/14 = 78.6% (49.2–95.3) | 12/13 = 92.3% (64.0–99.8) |
| STR[e] resistance | 10/15 = 66.7% (38.4–88.2) | 5/5 = 100% (47.8–100) |

[a]AST, antimicrobial susceptibility testing by culture; FQ, fluoroquinolones; SLI, second-line injectable; EMB, ethambutol; STR, streptomycin.
[b]Resistance to FQ was defined when any of the gyrA mutations were present or gyrA wild type was missing.
[c]Resistance to SLI was defined when any of the rrs mutations were present or rrs wild type was missing.
[d]Resistance to EMB was defined when any of the embB mutations were present or embB wild type was missing.
[e]Resistance to STR was defined when any of the rpsl/rrs mutations were present or rpsl/rrs wild type was missing.

sensitivity of GenID LPA RIF/INH to detect *Mtb* DNA in patients with undetectable AFB on smear microscopy, where the test cannot be recommended. A meta-analysis using four data sets with the first generation of LPA (Genotype MTBDR*plus* VER 1.0) and one data set with the second generation (Genotype MTBDR*plus* VER 2.0) reported 44.4% sensitivity in patients with undetectable AFB and 94.4% in patients with detectable AFB on sputum smear microscopy, compared to MGIT 960 culture, leading to a similar conclusion (4).

Sensitivity of the GenID LPA RIF/INH strip to detect *Mtb* DNA is higher in HIV-negative than in HIV-positive patients, which is likely due to the reduced inflammation and tissue destruction in HIV-positive TB patients, consequently reflected by less AFB detected on sputum microscopy (14).

Detection of MDR/RR-TB is the primary reason for using the GenID RIF/INH strip in clinical practice (15). A sensitivity of 84% for RIF and 90% for INH resistance showed inferior performance compared to Genotype MTBDR*plus* VER 1.0 and VER 2.0 and Nipro NTM+MDRTB LPA for RIF, where the sensitivity for the detection of RIF and INH resistance was 96% and 89% in a meta-analysis, respectively (4). Previous evaluations of the GenID RIF/INH strip report a sensitivity of 98% and 100% for RIF and INH resistance detection and a specificity of 100% for both drugs respectively; however, samples were collected retrospectively and were mostly from patients with detectable AFB on sputum microscopy (8). A specificity of 100% in all three strips was also reported in another study from Switzerland and South Africa, where stored sputum samples from patients with detectable AFB on sputum smear microscopy were evaluated by GenID LPA (7). In our prospectively evaluated cohort in a clinical setting in Namibia, specificity of the GenID LPA was 100% for RIF and 91.7% for INH, thus lower for INH than in the two other studies and the meta-analysis (4, 7, 8).

Detection of FQ resistance is of particular interest for the design of anti-TB drug regimens in cases of RIF resistance. The sensitivity of 86% and specificity of 95% of the GenID FQ LPA test from sputum are in line with the Cochrane review data, reporting

**TABLE 4** Proportion of samples that needed more than one GenID LPA run in order to obtain a valid result and proportion of samples without valid test result among smear positive and smear negative samples[a]

| Smear status | RIF/INH module | FQ/EMB module | STR/SLI module |
|---|---|---|---|
| Smear + and − | | | |
| Repeat testing[b] | 31.3 | 53.7 | 45.2 |
| Invalid results[c] | 0 | 1.8 | 23.6 |
| Smear + | | | |
| Repeat testing[b] | 45.9 | 60.5 | 48.0 |
| Invalid results[c] | 0 | 0 | 34.2 |
| Smear − | | | |
| Repeat testing[b] | 19.2 | 37.5 | 41.2 |
| Invalid results[c] | 0 | 5.8 | 0 |

[a]*Mycobacterium tuberculosis* detection and RIF/INH resistance on GenID LPA RIF/INH was tested in cohort 1 and cohort 2 and FQ/EMB and SLI resistance only in cohort 2. All numbers are percentages.
[b]Proportion of samples (%) that required >1 LPA run to obtain valid result.
[c]Proportion samples (%) where no valid result could be obtained.

82% and 99% in 9 studies with 1,771 samples from patients with detectable AFB on the sputum smear microscopy using Genotype MTBDR*sl* VER1.0 (16). SLI with the exception of AM is not recommended any more but was still a major component of MDR/RR-TB treatment during the time of this study. The GenID SLI strip demonstrated 75% sensitivity and 88% specificity compared to 87% and 100% in 8 studies including 1,639 participants when using Genotype MTBDR*sl* VER 1.0 (16).

In this study, GenID LPA testing was performed in a newly established laboratory after comprehensive training of the laboratory team, mirroring a real-world situation regarding the scarcity and need for establishing molecular TB diagnostics in African settings. Patient inclusion started only after several months of training and a pilot phase. Nevertheless, we frequently had to repeat tests in order to obtain valid test results. Despite the opportunity to rerun in particular the GenID SLI LPA, it showed invalid results (23%) mostly due to incomplete band formation. We cannot exclude that the high proportion of HIV-positive patients (73% in the cohort 1 and 49% in the cohort 2) may have affected test performance.

The study has a number of limitations. The sample size in cohort 1 and cohort 2 was small, which resulted in rather large confidence intervals. In order to allow a meaningful stratification for smear results and HIV status, results for *Mtb* detection and RIF and INH resistance were reported combined for cohort 1 and 2 (Table 2) but also separate for both cohorts (Table S1). In particular resistance to FQ and SLI was rare and results need to be interpreted with caution. For GenID RIF/INH, we used Xpert MTB/RIF resistance detection as reference standard for RIF and AST for INH. This decision was taken (1) based on data from a meta-analysis (17, 18), and (2) as it is known from later, yet unpublished work, that Xpert MTB/RIF predicted RIF resistance more accurately in this study than AST using a critical concentration of 1.0 mg/L for RIF instead of the later adopted 0.5 mg/L (19). Unfortunately, it was not possible neither to verify phenotypic AST results with genome sequencing of the specimens in question nor to increase sample size due to logistic and funding limitations. The impact of different *Mtb* lineages on *Mtb* detection could not be evaluated. The interpretation of the strips was done by two readers, but the sometimes faint band formation limits the accuracy of readings even among experienced strip readers. The high frequency of repeat testing to obtain valid results could be associated with a higher risk of errors during the laboratory processing of samples.

In conclusion, GenID LPA RIF/INH and FQ showed only acceptable diagnostic accuracy in an African, high-TB/HIV-incidence setting in patients with detectable AFB on the sputum microscopy, while the test performance was insufficient for the evaluation of sputum samples for patients with undectable AFB on sputum smear microscopy. The assay misses the target product profiles for drug-resistance testing, suggesting a minimal sensitivity above 95% for RIF and above 90% for INH, FQ, and SLI and specificity higher than 95% compared to culture-based AST, as recommended by WHO (13).

## MATERIALS AND METHODS

**Study design.** Following written informed consent, study participants were prospectively included into two cohorts between April and December 2017. In cohort 1, patients admitted to Katutura State Hospital Windhoek respiratory ward, presenting with symptoms of TB (fever, cough, nights sweats, or weight loss according to the Namibian TB diagnostic algorithm [12, 20]), were recruited every Monday during the study period. A sputum sample was tested for presence of *Mtb*, RIF, and INH resistance, using GenID LPA. Patients with detectable *Mtb* DNA from sputum specimen by Xpert MTB/RIF and/or detectable growth of *Mtb* in culture were diagnosed with active TB, as comparator group for GenID LPA RIF/INH test results.

In cohort 2, TB patients admitted to the MDR-TB ward at Katutura State TB hospital in Windhoek were recruited when Xpert MTB/RIF performed on sputum specimen was positive for RR-TB and/or results of *Mtb* AST performed on cultured organisms confirmed RIF resistance.

GenID LPA RIF/INH, FQ, and SLI from sputum for first-line and second-line *Mtb* drug resistance (in case of documented RIF resistance in first-line LPA) was performed. Results were compared to Xpert MTB/RIF for RIF resistance and to *Mtb* culture (Bactec MGIT 960; Becton Dickinson, Sparks, MD, USA) and phenotypic AST results for all other drugs. Unfortunately, genome sequencing was not available in this cohort as comparator.

**Laboratory procedures.** Sputum was treated with *N*-acetyl-ʟ-cysteine-sodium hydroxide for decontamination before further processing (21). Sputum smear microscopy, using auramine staining, TB

culture, and, if the culture was positive, phenotypic AST for RIF, INH, EMB, and STR in cohort 1 and in addition for FQ and SLI in cohort 2, was performed at the Namibia Institute of Pathology (NIP). Critical concentrations for AST were as follows: 1.0 mg/l RIF, 0.4 mg/l INH, 5.0 mg/l EMB, 0.25 mg/l moxifloxacin, 1.0 mg/l levofloxacin, 1.0 mg/l AM, 1.0 mg/l STR, 2.5 mg/l CM, and 2.5 mg/l KM. Sputum cultures were declared as "no growth" when they were not flagged positive by the MGIT culture device after 6 weeks of incubation.

GenID RBD2185/86 test kits (donated by the manufacturer) were used for *Mtb* detection and RIF and INH resistance testing and RBD2187/88 for FQ and EMB and RBD2184/89 for SLI/STR resistance detection. LPA testing by GenID LPA was performed at the research laboratory at University of Namibia School of Medicine in Windhoek.

GenID LPA testing was performed according to the manufacturer's instructions. DNA was extracted from decontaminated sputum samples, after aliquots were first used for smear microscopy and Xpert MTB/RIF and culture, using the hot sodium hydroxide and tris (HotSHOT) protocol and heat inactivation at 95°C for 5 min (22). A HotStar *Taq* DNA polymerase (Qiagen, Germantown, MD, USA) was used for DNA amplification in a GeneAmp PCR System 9700 thermocycler (Applied Biosystems, Waltham, MA, USA).

In case the GenID LPA showed an invalid result in the first test run, LPA was repeated, provided sufficient sputum was available. A maximum of two sputum samples were collected from each patient, and a maximum of four repeat runs of the GenID LPA were performed in order to obtain a valid result.

**Results interpretation.** Results reading and interpretation were done by two laboratory analyst according to the manufacturer's instructions. Results were declared invalid, if, with the available quantity of sputum DNA, no interpretable strip result was obtained or if the conjugate control and/or amplification control bands were not developed. The *Mycobacterium* universal control band was not included in this interpretation, which was stipulated by the manufacturer. *Mtb* detection was assessed using the *Mtb* band of the GenID RIF/INH strip. If RIF resistance was present, the GenID assays for FQ und SLI were performed.

Drug resistance was defined by a visibility of mutation bands and/or missing corresponding wild-type bands. Thus, if both wild-type and mutation bands were missing, this was also interpreted as drug resistance. The development of both a mutation band and a wild-type band for the same gene locus was considered heteroresistance and interpreted as resistance.

**Data collection, statistical analysis, and ethics.** Baseline demographic and clinical data were collected in patient interviews and by reviewing of patient treatment cards and recorded using the "Koch6" (Medicins sans Frontieres, Paris, France) database program. Routine laboratory data were captured from the Meditech laboratory database of NIP. Statistical analysis was performed with Stata/SE (version 16; Stata Corp, College Station, TX, USA).

The study protocol was approved by the Ethics Board of the Ministry of Health and Social Services in Windhoek, Namibia (Ref. No. 17.3.3).

## SUPPLEMENTAL MATERIAL

Supplemental material is available online only.

**SUPPLEMENTAL FILE 1**, PDF file, 0.1 MB.

## ACKNOWLEDGMENTS

We acknowledge the valuable input of S. Niemann and T. Niemann (Research Center Borstel) in improvement of the laboratory workflow. We also acknowledge the support in logistics, test kits, initial lab establishment, and training by AID (G. Schöllhorn, R. Preyer).

The authors report no conflict of interest.

The study was supported by an unrestrictional educational grant from AID to C.L. AID did not have access to the data, did not participate in the evaluation of the data, did not participate in the writing of the manuscript, and did not influence the decision to submit the manuscript for publication.

G.G. and C.L. designed the study; G.G. and E.N. implemented and supervised the study; E.S. and A.R. did the statistical analysis; H.E. collected and managed the clinical data; N.M. and A.D. did the laboratory work; G.G. and E.N. supervised the work; G.G. wrote the manuscript with input from E.S., A.R., E.N., and C.L. All authors reviewed and approved the final version of the manuscript.

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
