## [Reviewer comments · Microbiology Spectrum]

Microbiology Spectrum

Clinical evaluation of a line probe assay for tuberculosis detection and drug-resistance prediction in Namibia

Gunar Günther, Elmar Saathoff, Andrea Rachow, Azaria Diergaardt, Hilya Ekandjo, Nicola Marais, Christoph Lange, and Emmanuel Nepolo

Corresponding Author(s): Gunar Günther, Inselspital Bern

Review Timeline:

Submission Date:	April 4, 2022
Editorial Decision:	April 29, 2022
Revision Received:	May 3, 2022
Accepted:	May 16, 2022

Editor: Rosemary She

Reviewer(s): The reviewers have opted to remain anonymous.

Transaction Report:

DOI: <https://doi.org/10.1128/spectrum.00259-22>

Manuscript: Clinical evaluation of a line probe assay for tuberculosis detection and drug-resistance prediction in Namibia (Spectrum00259-22)

Comments to the reviewers:

Reviewer #1 (Major Comments for the Author):

The authors evaluated a 'novel' LPA for the detection of TB and drug-resistant TB and results were compared against Xpert and/or culture

Comment 1: Line 196 '...maximum of two sputum samples were collected from each patient' - Does this mean Xpert was also performed on both sputum samples and culture, if eligible, as well? Since multiple sputum samples were collected, wouldn't it make more sense to report results on a sample basis rather than patient basis?

Answer 1: Thank you very much for the thoughtful comment. Xpert and culture were only performed on one specimen. Maximum a 2nd sputum sample was collected for LPA from an eligible patient, if the quantity of sputum was insufficient to obtain valid result in the LPA. We set a limit of max. 4 LPA runs to obtain a valid result. This problem is illustrated in table 4. As the aim of the evaluation was also to report a valid result for a particular patient in order make treatment decisions, we opted to report results based on individual patients.

Comment 2: Line 243 Sensitivity and specificity from cohort 1 and 2 should not be combined.

Answer 2: Thank you for the important comment. We discussed the issue before analysis in our team. Cost and logistic constraints only allowed the small sample size. As we were keen to highlight the test performance stratified by smear results and also HIV coinfection, which has not been presented in previous evaluations of the GenID test, we opted for the combination of the cohorts. However, we did not present the test performance separate for cohort 1 and 2. We do so now in table S1.

The separate analysis does not affect the main findings and the conclusions if this work.

Comment 3: According to Figure 1, the Xpert detected 1 Rif resistant sample in cohort 1, but LPA for AFB smear-negative and smear-positive samples did not detect Rif resistance. This discrepancy is not mentioned in the result section.

Answer 3: We included the sentence: *In cohort 1 with presumed TB one single patient had RIF-resistance in Xpert[®] MTB/RIF, but resistance was not confirmed by LPA and DST. In recent work, to be published, including also WGS in a different cohort, we found frequently the rpoB L430P mutation. I.e. this mutation is not specifically detect by a mutation band of the GenID LPA and frequently documented as phenotypically drug-susceptible with the critical concentration for RIF, used during the study.*

Comment 4: In cohort 1, there were 27 TB positive-either Xpert positive and/or culture positive. While the positivity rate of 34% is quite high, 27 positive samples are not sufficient to evaluate a test for detection.

Answer 4. Thank you for the comment:

In order to address this matter, we evaluated also TB detection in the cohort 2, which led to 82 TB positive samples, and consequently to combination of cohort 1 and 2. We also decided to present the data to add possibly knowledge about the test to the public domain, which is not available elsewhere, being aware, that the number of samples does not allow final conclusion about test performance. We stated this under the limitations. Unfortunately, due to funding restrictions, we could not include more patients for a longer period of time.

Comment 5: Line 268 According to Figure 1, 87 patients were enrolled, and ultimately, results from 79 patients were analyzed; however, 134 samples were analyzed with the LPA. This is very confusing.

Answer 5: Thank you for the comment, we apologize for the confusion and hope we can clarify this issue. In cohort 1 79 out of 87 patients had a valid LPA result, in cohort 2 55 out of 67. Both cohorts, together 134 patients were included in the performance evaluation of the GenID INH/RIF LPA, which tested for MTB detection, RIF and INH resistance.

We changed the wording to clarify this: **42/134 (31.8%) patient samples in cohort 1 and cohort 2, evaluated with the LPA INH/RIF test for Mtb detection, required repeated testing in order obtain a valid test result. We also added the sentence in the paragraph on cohort description: 8 patients in cohort 1 (no Xpert / culture n=3; no valid LPA n=5) and 12 in cohort 2 (no Xpert / culture n=1; no valid LPA n=11) were excluded.**

General comments:

Comment 6: When sensitivity and specificity percentage values are given, it is also important to provide the numerator and the denominator, and not just the percentage.

Thank you for your comment. We included numerators and denominators, where ever sensitivities and specificities are given throughout the document.

Comment 7: It is a pity that no Sanger sequencing, or even whole genome sequencing, was performed and used as the true comparator for drug resistance.

Thank you for the comment. We fully agree and would have liked to use sequencing as a true comparator. Unfortunately, the technical capacities and the funding restrictions at the time of study did not allow to apply sequencing even if it was desired by the authors. The study had to be performed in a resource limited context and helped to build the research capacity in the team. The same team is now capable of doing tNGS locally.

Reviewer #1 (Minor Comments for Author (Required)):

Comment 8: Line 79 Please change rifampicin to rifampin, as this name is preferred in the Journal of Clinical Microbiology, throughout the manuscript where appropriate.

Thank you, this was done.

Comment 9: Line 109 Bruker took over Hain Lifescience - is the name of the company still the same?

The website still uses in the impressum the name: Hain Lifescience (<https://www.hain-lifescience.de/unternehmen/impressum.html>)

Comment 10: MTBDRplus - 'plus' is in italics. Please check throughout the manuscript, where appropriate.

Thank you, this was done.

Comment 11: Line 113 Please use the term 'antimicrobial susceptibility testing (AST)' instead of 'drug susceptibility testing (DST)' throughout the manuscript, where appropriate.

Thank you, this was changed.

Comment 12: Line 114 '...extrapulmonary and pediatric specimens' - plural "s"

Done

Comment 13: Line 117 '...same technology' - what is meant by this term?

Comment 14: We removed the word same – now it ready clearer:which is based on the technology of NAAT and in situ hybridization, detecting *Mtb* drug-resistance mutations against RIF, INH, EMB, FQ and SLI.

Comment 15: Line 126/127 Instead of 'concur with', should this be 'confer'?

Thank you for your comment. We changed wording of the sentence. It reads now:

Mutations at positions rrs 1401, 1402 are associated with resistance to AM/KM and mutations at position rrs 1484 with AM/KM/CM resistance.(6)

Comment 16: Line 173/174 reference is needed

Reference was included.

Comment 17: Line 176/177 phenotypic AST - please provide the concentrations tested for each compound in cohort 1 and for cohort 2; furthermore, it is not clear what method was used at NIP.

The following sentence was added:

Critical concentrations for AST were: RIF 1.0 µg/ml , INH 0.4 µg/ml, PZA 5.0 µg/ml, EMB 100 µg/ml, MXF 0.25 µg/ml, LFX 1.0 µg/ml, AM 1.0 µg/ml, STR 1.0 µg/ml CM 2.5 µg/ml, KM 2.5 µg/ml.

The method for AST was specified: It reads now.

Results were compared to Xpert[®] MTB/RIF, *Mtb* culture (BACTEC[®] MGIT 960; Becton Dickinson, Sparks, USA) and phenotypic AST results.

Comment 18: Line 197 '...maximum of four repeat runs' is unsustainable in a clinical laboratory.

We fully agree with the reviewer, that this is not sustainable. During the establishment of laboratory and pilot phase of study we noted the problems expressed in the paper regarding test performance. Table 4 quantifies the need for repeat testing. We opted for this approach of evaluating and depicting the challenges of test performance versus the mean number of test run to obtain a valid result. The maximum mean number of tests was 2.15 for obtaining a valid result on the strip for SLI.

Comment 19: Line 201 'technicians' - recommend using a broader term such as analyst. In some countries the term 'technician' is a well-defined human resources category and is differentiated from 'technologist'.

Thank you, this was changed.

Comment 20: Line 218 MSF is 'Medecins Sans Frontieres'

Thank you, this was changed

Comment 21: Line 280 One cannot consider the GenID LPA as novel since two studies were already published in 2014 and 2015.

We agree and removed the word novel in this line and "new" in the title of the paper.

Comment 22: Line 286 'paucibacillary' is not the opposite of AFB smear-positive...

We agree, that the word is not appropriate here and changed the sentence:

While the test sensitivity was 93% in patients with detectable AFB on sputum smear microscopy, test sensitivity was only 23% in patients without AFB in sputum smear microscopy.

Comment 23: Line 298 Reference #12 analyzed several different LPA - please double check the figures for MTBDRplus.

We checked the reference again: only four studies reported results on MTB detection with LPA Hain MTBDRplus V1 and one study with Hain MTBDRplus V2. The sensitivity for AFB smear positive and negative samples remain as reported. We rephrased the sentence. It reads now: A meta-analysis using four datasets with the first generation of LPA (Genotype[®] MTBDRplus VER 1.0) and one dataset with the second generation (Genotype[®] MTBDRplus VER 2.0) reports 44.4% sensitivity in patients with undetectable AFB and 94.4% in patients with detectable AFB on sputum smear microscopy, compared to MGIT 960 culture, leading to a similar conclusion.

Comment 24: Line 300-303 A reference is needed for this statement.

We added a reference, underlining this statement: Chamie G, Luetkemeyer A, Walusimbi-Nanteza M, Okwera A, Whalen CC, Mugerwa RD, Havlir DV, Charlebois ED. 2010. Significant variation in presentation of pulmonary tuberculosis across a high resolution of CD4 strata. *Int J Tuberc Lung Dis* 14:1295-302.

Comment 25: Line 329 MDRTBsl - "sl" is in italics

Thank you, we changed this.

Comment 26: Reference #2, #12, #13, and #15 are incomplete

Thank you, this was corrected.

Comment 27: Legend Figure 1 patients are not Xpert positive - samples are...

Thank you, we corrected the wording.

Reviewer #2 (Major Comments for the Author):

Thank you for giving me the opportunity to review the manuscript by Guenther and colleagues reporting on the performance of the AID suite of TB line probe assays in Namibia. The manuscript is well written and intelligible. I have some major and minor comments that the authors and editors may find helpful.

Major comments:

Comment 1: Title: My understanding is that the authors aimed to evaluate three (GenID RIF/INH, GenID SLI, GenID FQ) and not one product. This should be reflected in the study title.

Thank you for the comment. We evaluated the LPA, which to our understanding was one product, developed to detect a range of drug resistance mutations of Mtb, using 3 different strips under the same workflow and protocols. We aimed to use a short title for the study. But we adjusted the title based on your comments to:

Clinical evaluation of a line probe assay for tuberculosis detection and drug-resistance prediction in Namibia

Comment 2: Are the assays actually commercially available? I could not find the respective information on the manufacturer's website.

The assay is commercially available, has a CE mark and is in use by few laboratories for routine diagnostics. See link to the manufacturer webpage: <https://www.aid-diagnostika.com/kits/molekularbiologischer-assay/infektionsdiagnostik/antibiotika-resistenzen/tb-modul-inh-/-rif>

Comment 3: The role of SLI testing has significantly declined since recent changes to WHO therapeutic regimens and a push towards all oral regimens. KAN and CM are no longer recommended therapeutic agents. So the assay design appears somewhat outdated to me for a novel test. Is the test really "novel" (line 116)?

Thank you for the comment. We fully agree with this comment and removed the words novel and new from the manuscript.

Comment 4: The sample numbers (79 and 55) are low. What were the considerations around minimum sample numbers? Was there some sort of power calculation? Why did the authors choose not to run the study for a longer time?

The study was implemented in the context of building laboratory capacity and workflows for molecular diagnostics in a low resource setting. Capacity for molecular diagnostics of drug – resistance was not preexistent at the University of Namibia laboratories. Logistical challenges and funding constraints for staff and consumables allowed only a single center study and consecutive sampling for one year.

The authors would have liked to increase sample size, particularly for DR-TB patients. But for the reasons given, mostly funding, it was not possible to increase the sample size.

Comment 5: Were leftover samples used? Were they analyzed retrospectively? If so, were they frozen (at what temperature?) between routine testing and performance of the LPA?

No leftover samples were used. Samples were collected prospectively for the study and analyzed as fresh sputum samples in all cases with the LPA. No frozen samples were used.

Comment 6: Were operators performing the LPA blinded to the results of the routine diagnostic workup?

Operators were not blinded to routine workup. But the LPA laboratory had no direct access to the laboratory information system of NIP, and so didn't the operators. Also LPA were interpreted immediately, when results were obtained, while culture-based resistance results could not influence this interpretation, as they were available only weeks later.

Comment 7: The authors should mention the intended use of the assays as per the package insert. Does the manufacturer actually claim that the test can be used as a screening PCR to identify patients with TB (as opposed to a reflex test following another screening PCR to identify MDR-TB)?

The package insert says, that the test can be used as test for presence of *M. tuberculosis* and resistance testing. Quote package insert: "This single kit assay by GenID® GmbH, Straßberg, which is based on a multiplex PCR followed by reverse hybridization using sequence-specific oligonucleotide probes (SSOP), enables accurate and rapid modular detection of *Mycobacteria of the Mycobacterium tuberculosis complex* and its resistances to Isoniazid and Rifampicin."

We added a sentence which specifies this: **The LPA can be used for the detection of *Mtb* complex and *Mtb* drug-resistance mutations against RIF, INH, EMB, FQ and SLI.**

Comment 8: The Introduction/Discussion could perhaps reflect this alongside the comparison to the WHO TPP.

Along the same lines: What are the intended materials claimed by the manufacturer? Sputum only? Respiratory samples only? Pulmonary and extrapulmonary samples? Raw primary specimens or also pre-treated primary specimens? Only cultures? If the authors investigated any use that is off-label as per the claims of the manufacturer, this should be mentioned.

The manufacturer of the test does not specify the materials to be used: The package insert only says: ".....using DNA isolated from an appropriate specimen."

Comment 9: For any discussion around the sensitivity of the assay, it would be necessary to understand whether a single-copy (e.g. *rpoB*) or a multi-copy (e.g. IS6110) target is used for detection of MTBC.

Unfortunately the manufacturer does not specify in its documentation if a single copy of multi copy target is used.

Comment 10: In relation to my previous comment: Which TB lineages were covered by the sample panel (IS6110 copy numbers are variable between MTBC members)? Can the authors comment on whether the assay will show similar performance characteristics in settings with different regional epidemiology? If not, this should be mentioned as a limitation.

The manufacturer only states in the package insert, that the assay detects *M. tuberculosis complex*. Only two studies with the assay, performed in Europe with samples from Spain, Switzerland and South Africa have been performed. The studies focus on resistance detection. Therefore so far nothing is known about the performance of the LPA in different epidemiological settings. We added this point to the limitations. It reads: **The impact of different *Mtb* lineages on *Mtb* detection could not be evaluated.**

Comment 11: Lines 112-114: "LPA diagnostic performance is usually compared with culture-based drug susceptibility testing (DST), which remains the reference standard of Mtb resistance testing." This statement does not reflect the best practice. While culture can be considered the reference standard for detection of MTBC nucleic acid, predictions for RIF and INH susceptibility / resistance should not be compared with pDST alone but with a composite reference standard comprising pDST and the rapid molecular test that is validated in the respective routine diagnostic workflow (often Xpert or HAIN GenoType MTBDRplus or both). Discrepant samples should then be resolved using another method, preferably whole genome sequencing. This is important in the light of mutations conferring borderline resistance to rifampicin, which are not necessarily detected by pDST alone.

Thank you for this important statement: We fully agree with your comment. Ideally we would have wished to perform this study and compare LPA results with genome sequencing results. But logistics, setting and funding did not allow the use of sequencing methodology.

We had to battle in this study with the challenges of a resource limited setting. Stock outs of Xpert and culture reagent in the Namibia Institute of Pathology, who are doing Xpert MTB /RIF and culture/DST had impact on the study. Therefore we used for Mtb detection a combined reference standard of culture and Xpert MTB/RIF. Not doing so, would have further reduced the already low sample size.

A follow up study, using genome sequencing, currently under manuscript preparation highlights the discrepancies in resistance detection between Xpert MTB/RIF and culture in particular for rifampicin in Namibia. The now implemented change in the critical concentration for rifampicin also in the NIP laboratory should mitigate this problem.

The results of follow up study show, that Xpert MTB/RIF predicts more accurately rifampicin resistance, than phenotypic AST at the time of the GenID LPA study.

Therefore, based on this recent analysis we decided in the revised version of the manuscript, that reference standard for rifampicin resistance should be Xpert MTB /RIF, not phenotypic DST.

Table 2 and the results section were revised accordingly.

We added a sentence clarifying the desired reference standards for LPA performance evaluation:
LPA diagnostic performance for Mtb detection is mostly compared with Mtb culture.(3)
Antimicrobial susceptibility testing (AST) performance is compared increasingly to molecular (i.e. next generation genome sequencing and culture based methods as reference standards.(4)

Comment 12: References are partially outdated. For example, when relating to WHO guidance, the introduction should summarize the rapid molecular WHO-recommended tests as outlined in the WHO consolidated guidelines/operational handbooks on tuberculosis. Module 3: Diagnosis - Rapid diagnostics for tuberculosis detection 2021 update. These documents should be cited as references.

Thank you for the important comment. The updated guidelines for TB diagnostics from 2021 have been included in the introduction and references, specifying the current role of LPA in the TB diagnostic pathway. We did not discuss the PZA- LPA, as PZA is not test in the GenID assay.

Currently WHO recommends the use of LPAs in sputum-smear positive patients and from culture positive specimen as initial test for detection of resistance to RIF and INH instead of culture based phenotypic antimicrobial susceptibility testing (AST). In addition, LPAs are recommended in patients with multidrug-resistant (MDR)/ RIF-resistant (RR) TB as initial test to detect resistance to FQ and SLI.(2)

Comment 13: Table 2: The same reference standard (Xpert or culture) should be applied to all samples that are compared. In my view, for TB detection, culture should be used as reference and test performance characteristics for Xpert and the AID LPA should be compared head-to-head against culture.

Thank you for the important comment. We discussed the point during manuscript preparation. We combined the reference standard Xpert MTB/ Rif and culture for Mtb detection, as the samples size of the study is already low, and using only one reference standard would have reduced it further. Unfortunately the Namibia Institute of Pathology laboratory – the only culture lab in the country had stock outs of reagents for culture which forced us in this suboptimal

situation, as we could not use culture consistently as reference standard. This is the bitter reality of building research and research capacity in such settings. We aimed for culture as reference standard in all cases.

We calculated again, how many patients we would lose by using only culture as reference standard, not combined with Xpert for MTB detection:

16/79 in cohort 1 have no culture result, 6/55 in cohort 2 had no culture result.

Based on this calculations and the small sample size we still agree with the reviewer comments, but feel the combined reference standard is an acceptable alternative, considering the circumstances.

In addition we might argue, that recent diagnostic performance studies, like the evaluation of the Xpert MTB/XDR assay ([https://doi.org/10.1016/s1473-3099\(21\)00452-7](https://doi.org/10.1016/s1473-3099(21)00452-7)) define Mtb detection by use of Xpert MTB/Rif or Xpert Ultra.

Comment 14: For any comparison between the LPA and other rapid molecular tests, the authors should indicate the sample volumes that were used as an input and the range of volumes recommended by the manufacturer. In similar studies, the available leftover sample volumes used for the index test tend to result in input volumes that are rather on the lower end of the recommended range, whereas the volume used for the standard-of-care assay (e.g. Xpert) is typically rather on the higher end of the range. This may introduce systematic bias.

We agree with your comment. The manufacturer of GenID does not recommend a specific sample volume. In this study, a spot sample and early morning sample was used for Xpert MTB/RIF, smear microscopy and culture at the NIP laboratory. If sufficient sputum remained, DNA extraction was performed for LPA. If not, an additional sample was collected. This might explain partially the need for repeat testing, as depicted in table 4. We clarified this by editing the methods section with the sentence:

GenID[®] LPA testing was performed according to manufacturer instructions. DNA was extracted from decontaminated sputum samples, **after aliquots were used for smear microscopy, Xpert[®] MTB/RIF and culture first**, using the hot sodium hydroxide and tris (HotSHOT) protocol and heat inactivation at 95°C for 5 min.(15)

Comment 15: Line 358/359: To make it easier for the reader, the authors could briefly explain the criteria used in the WHO TPP.

Thank you for the comment. We added the TTP requirements. The sentence reads now:
The assay misses the target product profiles for drug-resistance testing, **suggesting a minimal sensitivity > 95% for RIF and > 90% for INH, FQ and SLI and specificity > 95% compared to culture-based AST**, recommended by WHO.(19)

Reviewer #2 (Minor Comments for Author (Required)):

Comment 16: Methods: Which critical concentrations were used for phenotypic testing?

The critical concentrations were added: **Critical concentrations for AST were: RIF 1.0 µg/ml , INH 0.4 µg/ml, PZA 5.0 µg/ml, EMB 100 µg/ml, MXF 0.25 µg/ml, LFX 1.0 µg/ml, AM 1.0 µg/ml, STR 1.0 µg/ml CM 2.5 µg/ml, KM 2.5 µg/ml.**

Comment 17: Lines 284-289: Redundant, could be shortened

Thank you for the comment. We completely changed the paragraph, based on the change of the reference standard for RIF – resistance from culture/DST to Xpert MTB/RIF, **avoiding redundant results.**

Comment 18: Line 303: Detection OF ...

Thank you, this was corrected

Comment 19: Figure 1 appears to have not been included in the PDF for review

We are sorry, the figure was uploaded.

Reviewer #3 (Major Comments for the Author):

The authors describe the results of an evaluation of the performance of the GenID LPA for the detection of Mtb and the detection of drug resistance. Although the scientific approach and experimental design are appropriate, the description of the results is a bit confusing.

Comment 1: For example, this reviewer found the information in Table 2, Figure 1, and lines 253-255 confusing and contradictory. Table 2 states a sensitivity of 90.5% (19/21) and a specificity of 82.8% (24/29) for detection of RIF resistance whereas the data in Figure 1 suggest a sensitivity of 84% (32 of 38 RIF-resistant MTB+ samples in cohort 2 with LPA results) and a specificity of 100% (21/21 RIF-susceptible MTB+ samples in cohort 1 with LPA results). A 2-by-2 table comparing the GenID result for RIF resistance with the Xpert result for RIF resistance would be useful to the reader.

Thank you for your very important comment. Data in figure 1 were based on resistance detection to RIF based in Xpert MTB/RIF, while in table 2 data were based on phenotypic DST. This explains the difference.

We now changed in table 2 the reference standard for RIF resistance detection to Xpert MTB/RIF. At the time of the study the critical concentration for RIF resistance used based on WHO recommendations was 1.0 µg/dl. It is known from later work, that this missed a number of RIF – resistant cases, which were false negative in phenotypic DST, compared to Xpert MTB/RIF. The sensitivity for RIF – resistance testing is now only 84.2%, compared to 90.5%, if phenotypic DST for RIF is the reference standard.

Table S2 gives similar results, if cohort 1 and cohort 2 are analysed separately.

We added a sentence in the methods section: **Results were compared to Xpert® MTB/RIF for RIF resistance, to *Mtb* culture (BACTEC® MGIT 960; Becton Dickinson, Sparks, USA) and phenotypic AST results for all other drugs.**

And we highlighted also the different reference standards in the discussion: **For GenID RIF/INH we used Xpert MTB/RIF resistance detection as reference standard for RIF, and AST for INH. This decision was taken, as it is known from later, yet unpublished work, that Xpert MTB/RIF predicted RIF resistance more accurately than AST with a critical concentration of 1.0 µg/ml for RIF, used at the time of the study.**

Comment 2: Lines 243-249. The authors should clearly justify the rational for combining the data from cohort 1 and cohort 2 with respect to assessing the ability to detect Mtb. Combining the two cohorts generates a patient population in which 61% of the patients have bacteriologically confirmed TB - a proportion that is unlikely to represent the situation in a clinical setting.

Thank you for this important comment. As highlighted earlier in our comments, the authors discussed this approach at length at the analysis stage.

We opted for the combination of the 2 cohorts to allow for meaningful stratification of results by smear status and HIV, as this is first study assessing the GenID assay in a real world, HIV- high incidence setting. The primary reason in real world setting is the use of the LPA for resistance detection, not MTB detection. As MTB detection and resistance detection are linked closely, using the GenID LPA, and considering the intentions to stratify, we opted to combine the cohorts. In cohort 2 TB was confirmed by Xpert MTB/RIF in all but 2 cases. We opted to also use this cohort for performance assessment of the GenID RIF/INH strip for TB detection, knowing the limitation of a total of 61% patients with bacteriologically confirmed TB. Table S1 now shows the performance of GenID® LPA RIF/INH, comparing cohort 1, cohort 2 and both cohorts together.

Comment 3: On the other hand, combining the cohort 1 and cohort 2 data to assess the performance of the assay for detecting drug-resistance in bacteriologically confirmed patients is appropriate.

Thanks you for the comment. As the main clinical use of the GenID assay is resistance testing, not Mtb detection, we kept table 2 in the main manuscript and added table S1 in the supplement.

Comment 4: It would be very useful for the reader if the authors described separately the performance of the assay for detecting Mtb in persons with presumed TB and the performance of the assay for detecting drug-resistance in persons with bacteriologically confirmed TB (Xpert or culture positive).

We did so in the revised manuscript and added table S1. We also referred to it in the results section in the sections about Mtb detection where we added the sentence: **Similar results were observed in a separate analysis of cohort 1 and 2 (table S 1).**

In the section about resistance prediction we added the sentence: **Table 2 also shows the diagnostic sensitivities in HIV-positive vs. HIV-negative patients, while table S1 shows diagnostics sensitivities and specificities for RIF and INH testing separate for cohort 1 and 2.**

Reviewer #3 (Minor Comments for Author (Required)):

Comment 5: Abstract: include 95% confidence intervals in for the performance parameters.

This has been done, thank you.

Comment 6: Line 156-159. In cohort 1, were the patients sufficiently ill to warrant admission to the hospital or were the patients just evaluated at in the respiratory ward?

Symptoms led to presentation at the internal medicine admissions department of the hospital. If the assessment was, that work up for TB is required, the workup was performed on the respiratory admissions ward.

Comment 7: Lines 230 to 231. The authors should discuss the potential impact on the performance of the assay of the 9% of patients in cohort 1 and 17% of patients in cohort 2 who did not have valid results.

Thank you for the comment.

We added a sentence in the discussion about the reduced performance by the lack of valid LPA results, which reads: **In addition, the test performance was reduced by 5.7% samples in cohort 1 and 16.4%, in cohort 2, which were excluded from the analysis mostly as no valid LPA result for the GenID RIF/INH strip could be produced.**

April 29, 2022

Dr. Gunar Günther
Inselspital Bern
Respiratory Medicine
Freiburgstrasse 15
Bern 3010
Switzerland

Re: Spectrum00259-22 (**Clinical evaluation of a line probe assay for tuberculosis detection and drug-resistance prediction in Namibia**)

Dear Dr. Gunar Günther:

Thank you for submitting your manuscript to Microbiology Spectrum. As you will see your paper is very close to acceptance. Please modify the manuscript along the lines I have recommended. As these revisions are quite minor, I expect that you should be able to turn in the revised paper in less than 30 days, if not sooner. If your manuscript was reviewed, you will find the reviewers' comments below.

When submitting the revised version of your paper, please provide (1) point-by-point responses to the issues I raised in your cover letter, and (2) a PDF file that indicates the changes from the original submission (by highlighting or underlining the changes) as file type "Marked Up Manuscript - For Review Only". Please use this link to submit your revised manuscript. Detailed instructions on submitting your revised paper are below.

Link Not Available

Sincerely,

Rosemary She

Reviewer (Editor) comments:

The response to JCM reviewer comments was thorough and thoughtful. The only revision I am requesting is to better substantiate the use of Xpert MTB/Rif assay as the reference standard for rifampin resistance - marked up manuscript lines 385-388. It would be preferable to find a published source or guideline that can be referenced rather than citing one's own unpublished work.

Preparing Revision Guidelines

- point-by-point responses to the issues I raised in your cover letter
- Upload a compare copy of the manuscript (without figures) as a "Marked-Up Manuscript" file.
- Each figure must be uploaded as a separate file, and any multipanel figures must be assembled into one file.
- Manuscript: A .DOC version of the revised manuscript
- Figures: Editable, high-resolution, individual figure files are required at revision, TIFF or EPS files are preferred

Please return the manuscript within 60 days; if you cannot complete the modification within this time period, please contact me. If you do not wish to modify the manuscript and prefer to submit it to another journal, please notify me of your decision immediately so that the manuscript may be formally withdrawn from consideration by Microbiology Spectrum.

Manuscript: Clinical evaluation of a line probe assay for tuberculosis detection and drug-resistance prediction in Namibia (Spectrum00259-22)

Comments to the reviewers:

The response to JCM reviewer comments was thorough and thoughtful. The only revision I am requesting is to better substantiate the use of Xpert MTB/Rif assay as the reference standard for rifampin resistance - marked up manuscript lines 385-388. It would be preferable to find a published source or guideline that can be referenced rather than citing one's own unpublished work.

Thank you for your feedback and comments.

We added a comment and references from 2 Cochrane reviews, which assess the appropriateness of Xpert MTB/Rif as screening test for MTB detection and RIF-resistance versus culture and LPA to the sentence.

We also clarified the second part of the sentence. We noted during later sequencing work, that pAST missed RIF-resistance with L430P mutation, which led to the adjustment of the critical concentration by WHO 2021. We also added this reference.

The sentence reads now:

This decision was taken (a) based on data from a metaanalysis (20, 21), and (b) as it is known from later, yet unpublished work, that Xpert MTB/RIF predicted RIF resistance more accurately in this study than AST using a critical concentration of 1.0 mg/ml for RIF instead of the later adopted 0.5 mg/ml (22).

We corrected further one reference, which was incorreced placed in the discussion. The sentence reads now.

A sensitivity of 84% for RIF and 90% for INH-resistance, showed inferior performance compared to Genotype[®] MTBDR^{plus} VER 1.0, VER 2.0 and Nipro[®] NTM+MDRTB LPA for RIF, where the sensitivity for the detection of RIF and INH-resistance were 96% and 89% in a meta-analysis, respectively.(4)

May 3, 2022

Dr. Gunar Günther
Inselspital Bern
Respiratory Medicine
Freiburgstrasse 15
Bern 3010
Switzerland

Re: Spectrum00259-22R1 (**Clinical evaluation of a line probe assay for tuberculosis detection and drug-resistance prediction in Namibia**)

Dear Dr. Gunar Günther:

Your manuscript has been accepted, and I am forwarding it to the ASM Journals Department for publication. You will be notified when your proofs are ready to be viewed.

Sincerely,

Rosemary She
Editor, Microbiology Spectrum
